# Electrostatic Atomized Water Particles Induce Disease Resistance in Muskmelon (*Cucumis melo* L.) against Postharvest Fruit Rot Caused by *Fusarium incarnatum*

**DOI:** 10.3390/jof9070745

**Published:** 2023-07-13

**Authors:** Samak Kaewsuksaeng, Prisana Wonglom, Anurag Sunpapao

**Affiliations:** 1Department of Plant Science, Faculty of Technology and Community Development, Thaksin University, Phatthalung Campus, Phatthalung 93210, Thailand; samak@scholar.tsu.ac.th (S.K.); prisana.w@tsu.ac.th (P.W.); 2Agricultural Innovation and Management Division (Pest Management), Faculty of Natural Resources, Prince of Songkla University, Songkhla 90110, Thailand

**Keywords:** *Cucumis melo*, EAWPs, enzyme activity, gene expression, induced resistance, H_2_O_2_

## Abstract

The postharvest quality of muskmelon can be affected by fruit rot caused by the fungus *Fusarium incarnatum*, resulting in loss of quality. The utilization of electrostatic atomized water particles (EAWPs) in agriculture applications has been shown to induce disease resistance in plants. Therefore, in this study, we determined the effect of electrostatic atomized water particles (EAWPs) on the disease resistance of muskmelon fruits against postharvest fruit rot caused by *F. incarnatum*. EAWPs were applied to muskmelon fruits for 0, 30, 60, and 90 min. EAWP-treated muskmelon fruits were inoculated with *F. incarnatum*, and disease progress was measured. Quantitative reverse-transcription polymerase chain reaction (qRT-PCR) of the chitinase *(CmCHI)* and β-1,3-glucanase *(CmGLU*) genes of *Cucumis melo* (muskmelon) was performed for EAWP-treated and -untreated muskmelon fruits. The activities of cell-wall-degrading enzymes (CWDEs), chitinase, and β-1,3-glucanase were also assayed in EAWP-treated and -untreated muskmelon fruits. The results showed that disease progress was limited by EAWP treatment for 30 min prior to pathogen inoculation. Muskmelon fruits treated with EAWPs for 30 min showed an upregulation of CWDE genes, *CmCHI* and *CmGLU*, as observed by qRT-PCR, leading to high chitinase and β-1,3-glucanase activities, as observed through enzyme assays. The results of SEM microscopy revealed that the effect of the crude enzymes of EAWP-treated muskmelon caused morphological changes in *F. incarnatum* mycelia. Furthermore, treatment with EAWPs preserved postharvest quality in muskmelon, including with regard to texture stiffness and total chlorophyll contents, compared to untreated muskmelon. These results demonstrate that the pretreatment of muskmelon with EAWPs suppresses the development of *F. incarnatum* in the early stage of infection by regulating gene expression of CWDEs and elevating the activities of CWDEs, while also maintaining postharvest muskmelon quality.

## 1. Introduction

Muskmelon (*Cucumis melo* L.) is a species of *Cucumis* plants in the Cucurbitaceae family that has been bred into many cultivated varieties for commercial purposes worldwide. The nutritional value of muskmelon is high, because it contains ascorbic acid, carotene, folic acid, and potassium, along with other beneficial bioactive compounds for human health [1]. In Thailand, the cultivation of muskmelon has increased to meet market demand, with an annual production of about 16,440 ton (Department of Agriculture Extension). Muskmelon is commonly cultivated in Thailand in polyhouses to maintain a favorable environment and avoid insect pests. However, Thailand is located in tropical and subtropical regions, where the climate favors pathogen germination, so muskmelon production is challenged by several diseases during all stages of growth [2,3,4]. Postharvest fruit rot of muskmelon is one of the most destructive diseases affecting sealable muskmelon, and the causal agent was recently isolated and identified as *Fusarium incarnatum* [5]. *Fusarium* sp. normally infects plants through wounds during rain or irrigation [5]. The pathogen starts to invade plant tissues and rapidly spread to other parts of plants. The general morphological characteristics of *F. incarnatum* include a cotton-like colony and rapid growth on PDA. The hyphae of *F. incarnatum* are hyaline septate with a smooth cell wall [5]. Some *Fusarium* isolates produce cellulase and pectinase, which contribute to the weakening and successful invasion of the host plant [6], causing tissue rot. After harvesting, muskmelon fruits are normally stored at ambient temperature (28 ± 2 °C) before delivery to the market shelf. During the rainy season, about 10% of muskmelon fruits are infected with fruit rot and postharvest fruit rot [5]. The disease primarily starts from wounded or cracked muskmelon fruits near peduncles, which then become rotten with mycelia-covered lesions, and wet rotten tissues that can be observed beneath the lesions [5]. 

Plant defense mechanisms against pathogens play a crucial role in resisting invading pathogens. Plants monitor a broad range of defense mechanisms against abiotic and biotic stresses. Plants respond to abiotic and biotic stresses through oxidative burst of cells, synthesis of phytoalexin, and the production of pathogenesis-related proteins (PR) and producing and accumulating reactive oxygen species (ROS) [7,8,9]. PR proteins are accumulated, localized in the infected cells and neighboring tissues, in order to limit pathogen spread and improve the defensive capacity of plants [10]. Of the 17 different PR proteins, PR2 and PR3 function as glucanase and chitinase [11], and are involved in plant defense against fungal pathogen infection. Furthermore, ROS act as signaling molecules for regulating growth, development, and stress response [7]. ROS in plants include super oxide (O_2_^−^), hydrogen peroxide (H_2_O_2_), and hydroxyl (OH^−^) groups. H_2_O_2_ is one of the most studied ROS species, and is stable [12] and acts as a signaling ROS for plant development and stress responses [13]. However, despite extensive studies on plant defense mechanisms, the roles of these mechanisms are not fully understood. H_2_O_2_ is a major ROS that plays diverse roles in plant defense against pathogens [14]. 

Electrostatic atomized water particles (EAWPs) are the products of electrostatic atomization, and are produced and released in fine liquid droplets with a diameter ranging 10 to 100 μm [15]. EAWPs consist of ROS, including H_2_O_2_, O_2_^−^ and OH^−^, as well as nitric oxide [16,17,18]. EAWPs can delay cell senescence in fruits and vegetables [18,19], suppress chlorophyll breakdown and senescence in horticultural crops [20,21], and induce disease resistance in tomato against *Botrytis cinerea* [22]. With respect to postharvest quality, application of EAWPs for an appropriate period of time (90 min) has been shown to result in color and total chlorophyll content of a spear of asparagus being maintained during storage at 4 °C [21]. Application of EAWPs for 60 min was able to maintain the postharvest quality of ‘Namwa’ bananas by delaying ripening and senescence [19]. Furthermore, for disease control, the application of EAWP has been shown to induce disease resistance in tomato by upregulating the nitrate reductase gene and salicylic-dependent chitinase 3 [22]. The induction of H_2_O_2_ in transgenic plants enhances resistance to bacterial and fungal pathogens [23]. 

Thus, these studies led us to hypothesize that EAWPs containing H_2_O_2_ may induce a defense response in muskmelon plants against *F. incarnatum*. Postharvet fruit rot negatively impacts the quality and quantity of muskmelon fruit production, with a disease incidence of 10% on the market shelf. Pretreatment with EAWPs for an appropriate time could help to decrease disease development and maintain the postharvest quality of muskmelon fruit during storage. Therefore, in this study, we investigated the effect of applying EAWPs on disease resistance in muskmelon against postharvest fruit rot caused by *F. incarnatum*.

## 2. Materials and Methods

### 2.1. Plant Material and Pretreatment of Muskmelon with EAWPs

Muskmelon fruits in the harvesting stage (13–15 cm in diameter and weight of 0.8–1 kg) were obtained from Krisamai Plantation Farm, Hatyai, Songkhla, Thailand. Each muskmelon was surface disinfected with 70% ethanol prior to treatment with EAWPs. The muskmelon fruit was kept in a plastic box (24 × 35 × 18 cm) and exposed to EAWPs. Each muskmelon was treated constantly with EAWPs, generated using a Panasonic F-GMK01 Nano^e^ Air Purifier, for 0 (control), 30, 60, 90, or 120 min in a closed plastic box, as shown in Figure 1. The experiment was performed in 5 replications, and each replicate consisted of 2 fruits. The muskmelons were immediately subjected to pathogen inoculation after EAWP treatment.

### 2.2. Pathogen Inoculation and Disease Assessment

The postharvest fruit rot pathogen of muskmelon, *F. incarnatum*, which was isolated from infected muskmelon in previous study [5], was obtained from the Culture Collection of Pest Management, Faculty of Natural Resources, Prince of Songkla University, Thailand. The spore suspension was inoculated onto plant tissues following the method reported by Pornsuriya et al. [24]. The fungal pathogen was cultured on potato dextrose agar (PDA) and incubated at ambient temperature (28 ± 2 °C) for 7 days for sporulation. The conidia of *F. incarnatum* were collected, and their concentration was adjusted with sterile distilled water (DW) to 1 × 10^6^ conidia/mL. The treated muskmelons were wounded using fine needles (0.8 mm in diameter), and 10 µL of *F. incarnatum* spore suspension was applied onto the wounded muskmelon. The inoculated muskmelons were then incubated at ambient temperature for 3 days, and symptom development was observed. Lesion development was measured at 3 days post inoculation. 

### 2.3. Formation of H_2_O_2_ from EAWPs

To confirm that the closed atmosphere contained H_2_O_2_, quantitative H_2_O_2_ was measured under the same conditions described in Section 2.1. Petri dishes containing 3 mL of distilled water (DW) were exposed to the same EAWPs in a closed plastic box for 0 (control), 30, 60, 90, or 120 min, as described in Section 2.1. The accumulation of H_2_O_2_ in the DW was measured using the ferric thiocyanate method, with some modifications [25].

### 2.4. qRT-PCR of PR Protein Genes

Quantitative RT-PCR was conducted to observe the expression of PR proteins at the molecular level [26]. EAWP-treated muskmelon peels were subjected to RNA extraction using Trizol reagent (ThermoFisher, Waltham, MA, USA), and total RNA was dissolved in RNase-free DW and immediately used as an RNA template. The total RNA was reversed-transcribed to complementary DNA (cDNA) and subjected to quantitative reverse-transcription PCR (qRT-PCR), following the method of Dumhai et al. [27]. The qRT-PCR reaction mixtures contained 1 ng of cDNA as the template and iScript One-Step RT-PCR reagent with SYBR Green (Bio-Rad, Hercules, CA, USA). The internal reference gene in this study was actin (*act*), which we used to normalize the variations in the input total cDNA template between the control and EAWP-treated specimens. The primer pairs of PR protein genes (chitinase and β-1,3-glucanase) are shown in Table 1. The relative gene expression, as fold change in expression of PR protein genes relative to actin as the internal reference gene, was analyzed with Bio-Rad CFX Manager analysis software (Bio-Rad, Hercules, CA, USA).

### 2.5. Enzyme Assay

EAWP-treated muskmelon was subjected to an enzyme assay to determine defense-related and cell-wall-degrading enzymes [28,29]. Muskmelon peels were homogenized with potassium phosphate buffer (KPB) pH 7.4 and centrifuged at 12,000× *g* for 20 min. The supernatant was immediately used as the crude enzymes for the enzyme assay. The cell-wall-degrading enzymes used in this study included chitinase and β-1,3-glucanase. The activities of chitinase and β-1,3-glucanase were determined via the 3,5-dinitrosalicylic acid (DNS) method [30]. The chitinase activity was determined using 1% colloidal chitin as a substrate; glucose, a reducing sugar, as a product, was observed using a UV-5300 UV/VIS spectrophotometer at 575 nm. The β-1,3-glucanase activity was quantified using laminarin as the substrate, and the product N-acetyl-D-glucosamine was observed at 550 nm. 

### 2.6. Scanning Electron Microscopy Observation

To test the effect of the CWDEs produced from the muskmelon fruits treated with EAWPs, scanning electron microscopy was performed. EAWP-treated muskmelon peels were subjected to extract crude enzymes with potassium phosphate buffer pH 7.0, as described in Section 2.5. EAWP-untreated muskmelon fruits were used as the control group. Agar plugs of *F. incarnatum* were cut from the edge of 5-day-old colonies and incubated in crude extract at 37 °C for 1 h [31]. The samples were fixed in 3% glutaraldehyde at 4 °C overnight. The samples were then dehydrated in gradual alcohol series (30, 50, 70, 80, 90, and 100%) and dried to perform critical point drying (CPD). Finally, samples were coated with gold particles and observed using a scanning electron microscope (SEM, JSM-5800 LV, JEOL, MA, USA). Each treatment was composed of 5 replicates, and the experiment was conducted twice. 

### 2.7. Postharvest Quality Measurement

The EAWP-treated muskmelon at 0 min (control) and the most effective time from those listed in Section 2.1 were subjected to testing for postharvest quality. EAWP-treated muskmelons were stored at 20 °C from 0 to 9 days, and postharvest quality was tested at 3, 6, and 9 days. The parameters used to determine postharvest quality included percentage of weight loss, fruit stiffness, total soluble solid, and total chlorophyll content. Each treatment contained 3 muskmelon fruits (3 replicates), and the treatment was repeated twice. The percentage of weight loss was measured in EAWP-treated muskmelon after 3, 6, and 9 days of storage. Fruit firmness (stiffness) was measured using a System TA.Xtplus Texture Analyzer (Stable Micro Systems, Godalming, UK) and is expressed in Newtons (N). The total soluble solid content of the muskmelons was analyzed using a hand refractometer (N1; Atago Co., Ltd., Tokyo, Japan), and is expressed in degrees Brix in fresh muskmelon tissues. Total chlorophyll content was determined following the method of Kaewsuksaeng et al. [32]. 

### 2.8. Statistical Analysis

Experiments were conducted following a completely randomized design, and analysis of variance (ANOVA) was applied to the data. The data were subjected to an analysis of the significant differences among the treatments using Student’s *t*-test and Tukey’s test. 

## 3. Results

### 3.1. Application of EAWPs Limits Disease Progression

To confirm that EAWPs had been released from device, the accumulation of H_2_O_2_ in the Petri dishes was measured. The results showed that H_2_O_2_ accumulated in the Petri dishes containing DW from 30 to 120 min (Figure 2). 

The increase in H_2_O_2_ content was related to the duration of application. Compared with the untreated control muskmelon, pretreatment with EAWPs produced a significant reduction in lesion diameter at 3 days after challenge inoculation with *F. incarnatum*. Treatment with EAWPs for 30 min was the most effective at restricting the disease development of *F. incarnatum*, whereas the control group and other treatments showed more severe disease (Figure 3). We observed cotton-like mycelia of *F. incarnatum* and fruit rot beneath the fungal colony in the muskmelon (Figure 3). Muskmelon fruit pretreated with EAWPs for 30 min showed no fruit rot at the inoculation points and remained healthy in comparison to those in the other treatments. Therefore, the EAWP treatment for 30 min was considered the optimal condition, and was used for further study.

### 3.2. Overexpression of PR Protein Genes by EAWPs in Muskmelon

In this study, we determined expression levels of PR genes including *CmCHI* and *CmGLU,* which are expressed as chitinase and β-1,3-glucanase in qRT-PCR. Our results showed that the expression levels of *CmCHI* and *CmGLU* were significantly higher in muskmelon treated with EAWPs for 30 min than in the control (without EAWP treatment), as shown in Figure 4. The relative expression levels of *CmCHI* were 2.71 and 0.91 for the EAWP-treated muskmelon and untreated muskmelon, respectively. The relative expression levels of *CmGLU* were 3.07 and 0.85 for the EAWP-treated muskmelon and untreated muskmelon, respectively (Figure 4).

### 3.3. Effect of EAWPs on Defense-Related Enzymes and CWDEs’ Activity in Muskmelon

To determine the defense response owing to muskmelon pretreatment with EAWPs, the activities of CWDEs were determined via an enzyme assay. The enzyme activities of CWDEs in muskmelon fruit treated with EAWPs for 30 min were significantly higher than those in the control (Figure 5). The chitinase activity was 0.034 and 0.008 U/mL for treated and control muskmelon, respectively. The *β*-1,3-gucanase activity was 0.240 and 0.002 U/mL for treated and control muskmelon, respectively (Figure 5). 

### 3.4. Effect of EAWP-Treated Muskmelon Crude Enzymes on Morphological Changes in Fusarium incarnatum

To test the effect of the crude enzymes of EAWP-treated muskmelon on the morphological changes in *F. incarnatum*, SEM was used. SEM microscopy showed abnormal fungal cell walls and abnormal shapes for the *F. incarnatum* mycelia, as indicated by the wilting and rough mycelia; in comparison, the mycelia of the control remained healthy (Figure 6). 

### 3.5. Effect of EAWPs on Postharvest Quality

Treatment with EAWPs not only induced resistance in muskmelon against the postharvest fruit rot pathogen *F. incarnatum*, it also maintained postharvest quality (Figure 6). We found no statistical difference in weight loss between EAWP-treated muskmelon fruits and the control (Figure 7A). Fruit firmness, or fruit stiffness, was also measured, and the results showed that the fruit stiffness of the EAWP-treated muskmelon fruits was significantly higher than that of the control after 3, 6, and 9 days of storage (Figure 7B). The total soluble solid content fluctuated, and varied over time (Figure 7C). The chlorophyll content was also detected in the EAWP-treated muskmelon fruits to be significantly higher than that of the control (Figure 7D). 

## 4. Discussion

In this study, we tested the effectiveness of EAWP treatment, which generated H_2_O_2_, for inducing disease resistance in muskmelon against the postharvest fruit rot caused by *F. incarnatum*. Our results demonstrated that the application of EAWPs for 30 min was appropriate for limiting disease progress on inoculated muskmelon fruits (Figure 3). The upregulation of cell-wall-degrading enzymes (*CmCHI* and *CmGLU*) was associated with high activities of CWDEs (chitinase and β-1,3-glucanase), resulting in abnormal *F. incarnatum* morphology and disease resistance in muskmelon fruits (Figure 4, Figure 5 and Figure 6). Furthermore, the application of EAWPs maintained the postharvest quality of muskmelon fruits (Figure 7).

Infection with *Fusarium* species can cause fruit rot, dirty panicles, and wilting in several plant species [5,33,34]. Plants respond to fungal infection through the production of antimicrobial compounds and/or ROS [35]. H_2_O_2_ plays various important roles in plants, with multifaceted functions in plants’ defense against fungal invasion [14]. For instance, H_2_O_2_ exerts direct antimicrobial activity in the prevention of pathogen infection [36]. H_2_O_2_ may act as a signal transducer of systemic acquired resistance and induced gene expression [37,38]. In the present study, we found that muskmelon fruits pretreated with EAWPs showed enhance resistance against *F. incarnatum*, observed on the basis of symptom limitation (Figure 3) and the increased transcription levels of the defense-related genes *CmCHI* and *CmGLU* (Figure 4). We selected an EAWP application time of 30 min in order to investigate gene expression and enzyme activities in muskmelon fruits against *F. incarnatum* due to their capacity to limit lesion development (Figure 3). These results suggested that defense-related genes were induced in the muskmelon fruits treated with EAWPs for 30 min. Similar results was observed in tomato plants pretreated with EAWPs, where an upregulation of salicylic acid (SA)-dependent chitinase 3 (*chi3*) was observed after the leaves were inoculated with *Botrytis cinerea* [22]. Therefore, the application of EAWPs for 30 min is considered appropriate for inducing disease resistance against the postharvest fruit rot caused by *F. incarnatum* in muskmelon fruits.

Although the expression of PR genes (*CmCHI* and *CmGLU*) and the enzyme activity of chitinase and β-1,3-glucanase in EAWP-untreated muskmelon fruits demonstrated a significantly lower expression and activity level compared to EAWP-treated muskmelon. However, untreated muskmelon fruits showed expression of PR genes and activity of both enzymes. This phenomenon may be due to PR proteins being strongly induced by wounding or infection by pathogens, and it has been reported that wounding the plant tissues induces expression of PR genes in tobacco [39]. Furthermore, artificial wounding has been reported to enhance the expression level of PR-10 in bilberry leaf [40]. In our study, muskmelon fruits were wounded by needles before inoculation with *F. incarnatum* spore suspension and DW (control). This wounding may have induced expression of PR genes and enzyme activity, as observed in Figure 4 and Figure 5.

Another method for preventing fungal infection is via a mechanism that involves the production of CWDEs. The production of CWDEs, including chitinase and β-1,3-glucanase, in plants tissues can be caused by biotic and abiotic stresses [41,42,43], as well as by some antagonist applications to plants [44,45]. Both enzymes play a crucial role in hydrolyzing the fungal cell wall and limiting fungal invasion into plant tissues [46,47]. In the present study, pretreatment of muskmelon fruits with EAWPs for 30 min elevated the activities of both chitinase and β-1,3-glucanase in plant tissues compared with those of the untreated control fruits (Figure 5). The high levels of both enzymes limited fungal invasion into the plant tissues. Thus, pretreatment of EAWPs for 30 min is an appropriate for generating H_2_O_2_ with the aim of inducing high activities of both chitinase and β-1,3-glucanase in muskmelon fruits. This high activity of both enzymes may be related to the high transcriptional levels of *CmCHI* and *CmGLU* in muskmelon fruits, which were correlated with symptom suppression against postharvest fruit rot. Furthermore, the SEM results confirmed the activity of CWDEs produced by EAWP-treated muskmelons, which resulted in an abnormal morphology of *F. incarnatum* mycelia (Figure 6). This result suggested that the CWDEs produced by EAWP-treated muskmelons were involved in changing the morphology of the fungi. These findings are in agreement with those of Wonglom et al. [28], who reported that crude extracts from plants containing CWDEs are able to hydrolyze the fungal cell wall. 

As EAWPs contain ROS such as H_2_O_2_, these molecules may act directly on muskmelon cells, leading to changes in the expressions of the genes involved in the defense response. EAWPs have been applied in several plant species to prolong and maintain postharvest quality. The effects of H_2_O_2_, which is generated by EAWPs, on postharvest quality include the suppression of chlorophyll degradation, which results in the postharvest degreening of green sour citrus [18]. For instance, Salaemae et al. [19] applied EAWPs to prolong the postharvest quality of Namwa banana (Musa × paradisiaca). Kaewsuksaeng et al. [32] demonstrated in relation to the postharvest quality of mangosteen that application of EAWPs delayed the degradation of chlorophyll (Chl) and the activity of Chl-degrading enzymes in the calyx. EAWP treatment has been shown to maintain the color, chlorophyll content and firmness of asparagus [21]. Moreover, the application of EAWPs has been shown to successfully induce disease resistance in tomato against *Botrytis cinerea* [22]. Our findings show that pretreatment with EAWPs in muskmelon fruits not only induces disease resistance against postharvest fruit rot, but also allows some postharvest fruit qualities to be preserved. We found no difference in the weights of treated and untreated fruits, whereas the fruit firmness and chlorophyll content were higher in the EAWP-treated muskmelon than in the control (Figure 7). These findings suggest that the H_2_O_2_ generated by EAWPs results in fruit firmness and chlorophyll content being maintained in muskmelon fruits.

Based on the results of the present study, it can be concluded that pretreatment of muskmelon fruits using EAWPs for 30 min induces disease resistance against *F. incarnatum* and limits lesion development. In order to prevent muskmelon fruits from experiencing postharvest fruit rot, and in order to maintain postharvest quality, it is recommended that pretreatment with EAWPs for 30 min be applied to muskmelon after harvesting and/or before delivery to the market shelf. 

## 5. Conclusions

In our current study, EAWPs induced defense-related gene expression (*CmCHI* and *CmGLU*) and elevated the activities of chitinase and β-1,3-glucanase, thereby limiting fungal invasion into muskmelon fruits. Our findings suggest that EAWPs potentiate pathogen-induced defense gene expression and activate disease resistance against the postharvest fruit rot pathogen *F. incarnatum* in muskmelon fruits. Therefore, EAWPs may be able to be used to control postharvest disease in fruits and vegetables. 

## Figures and Tables

**Figure 1 jof-09-00745-f001:**
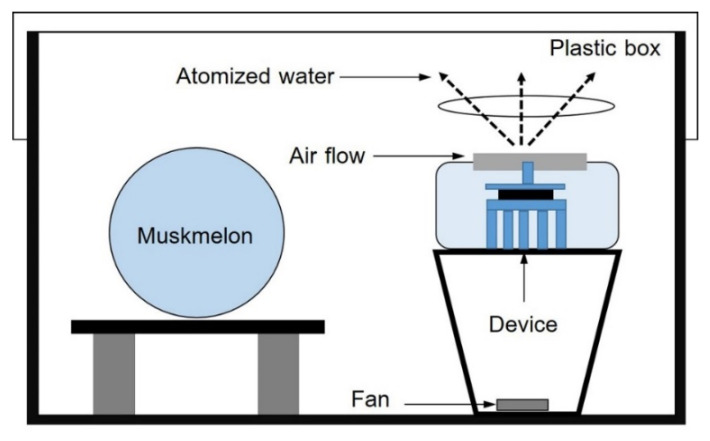
Schematic of apparatus used to generate air flow containing EAWPs for the treatment of muskmelon fruit in a closed chamber.

**Figure 2 jof-09-00745-f002:**
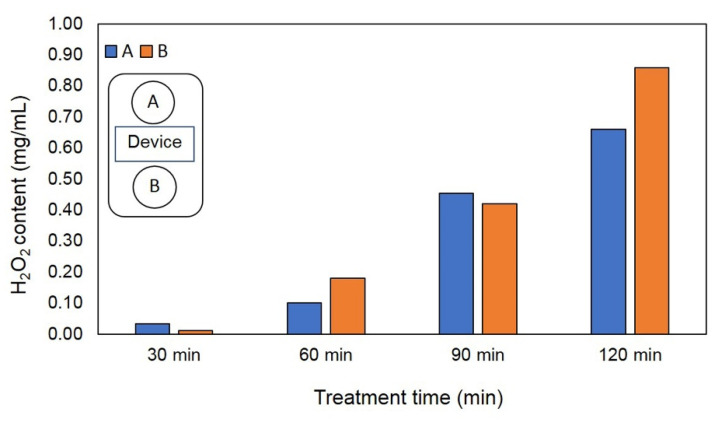
Electrostatic atomized water particle (EAWP)-induced hydrogen peroxide accumulation in Petri dishes (A, B) containing 3 mL of distilled water.

**Figure 3 jof-09-00745-f003:**
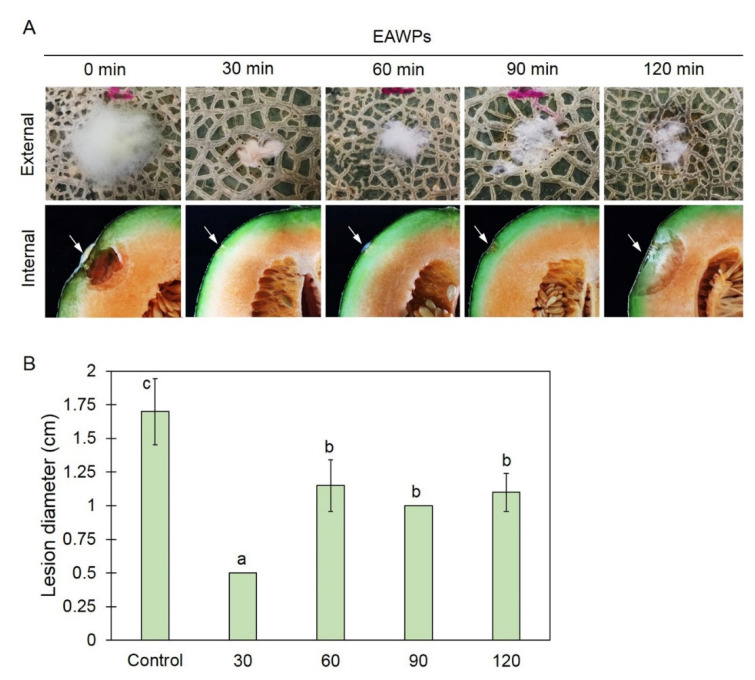
Effect of electrostatic atomized water particle (EAWP) treatment on postharvest fruit rot development on muskmelon. Muskmelon fruits were inoculated with *Fusarium incarnatum* after treatment with EAWPs for 0 (control), 30, 60, 90, or 120 min, and lesion development at 3 days post inoculation was evaluated. External and internal symptoms of muskmelon fruits (**A**) and lesion diameter (**B**). Error bars indicate standard deviation (SD) of application time (min.), whereas letters indicate significant difference among treatments according to Tukey’s test (*p* < 0.05).

**Figure 4 jof-09-00745-f004:**
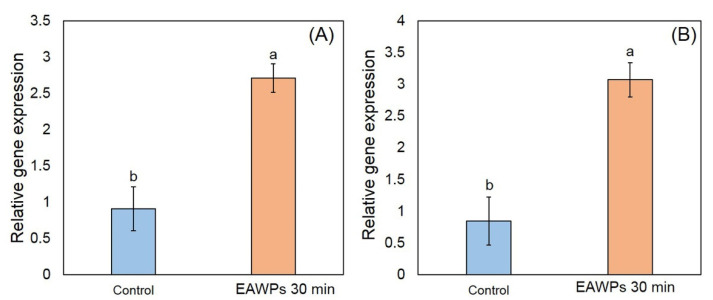
Relative gene expression of *CmCHI* (**A**) and *CmGLU* (**B**) in EAWP-treated muskmelon fruits. Letters indicate significant differences among treatments according to Tukey’s test (*p* < 0.05).

**Figure 5 jof-09-00745-f005:**
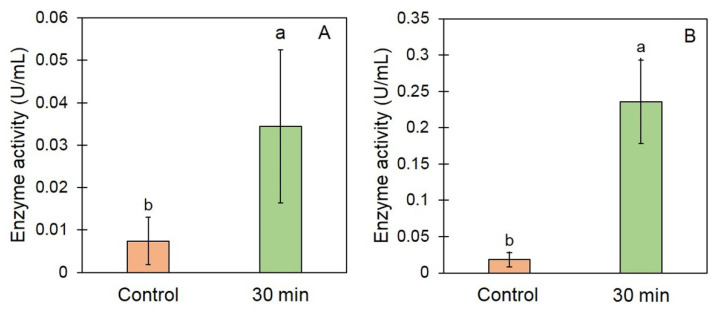
Effect of electrostatic atomized water particles on cell-wall-degrading enzyme activity: chitinase (**A**) and β-1,3-glucanase (**B**). Muskmelon fruits were inoculated with *Fusarium incarnatum* after treatment with EAWPs for 0 (control) and 30 min. Error bars indicate the standard deviation (SD) of enzyme activity (units), whereas letters indicate significant differences among treatments according to Tukey’s test (*p* < 0.05).

**Figure 6 jof-09-00745-f006:**
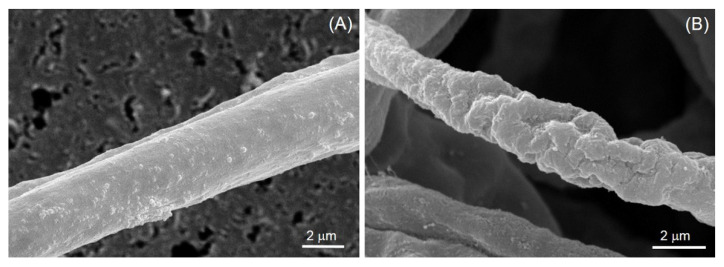
Scanning electron microscopy showing mycelia of *Fusarium incarnatum* treated with crude extract of EAWP-untreated muskmelon (**A**) and EAWP-treated muskmelon for 30 min (**B**).

**Figure 7 jof-09-00745-f007:**
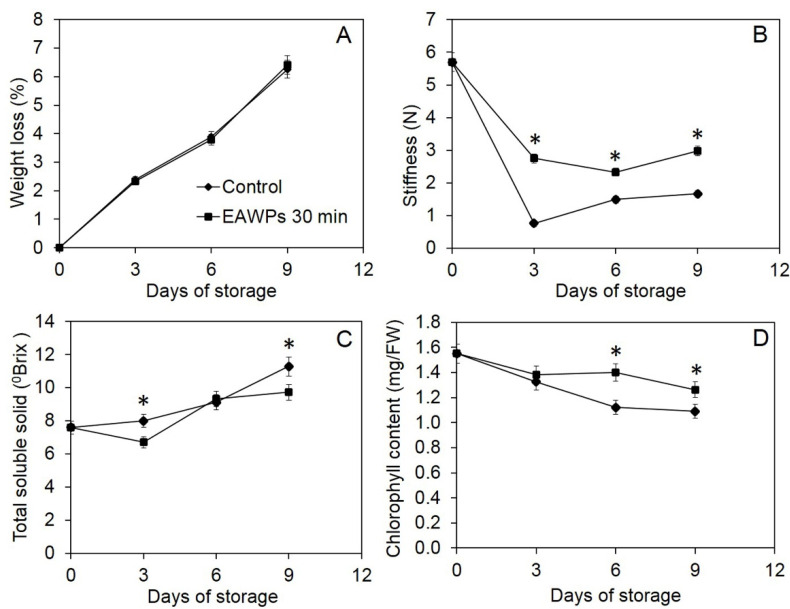
Postharvest quality of EAWP-treated muskmelon at 0 (control) and 30 min, including weight loss (**A**), fruit stiffness (**B**), total soluble solid content (**C**), and total chlorophyll content (**D**). Asterisks indicate significant differences between treatments according to Student’s *t*-test (*p* < 0.05).

**Table 1 jof-09-00745-t001:** Specific primer pairs for the gene expression determinations with quantitative real-time reverse-transcription polymerase chain reaction (qRT-PCR).

Gene	Accession No.	Primer	Sequence (5′→3′)	Product Size (bp)
*CmCHI* ^1^	AF241538	Chi-F	AGGATCCACAATGTCCAAGC	160
		Chi-R	CCGGTGGTTTGATGAGAAGT	
*CmGLU*	AF459794	Glu-F	AGAATGGTGGAGGATCGTTG	182
		Glu-R	GTCAGACATGGCGAACACAT	
*CmACT*	AB033599	ACT-F	TGTTGGTCGACCTCGTCATA	234
		ACT-R	GGGTTGAGTGGTGCTTCAGT	

^1^ *CmCHI*, chitinase; *CmGLU*, *β*-1,3-glucanase; *CmACT*, actin genes.

## Data Availability

Not applicable.

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
