# Peer review of "Electrostatic Atomized Water Particles Induce Disease Resistance in Muskmelon (Cucumis melo L.) against Postharvest Fruit Rot Caused by Fusarium incarnatum"

_jof, 2023, doi:10.3390/jof9070745_

Round 1

Reviewer 1 Report

This paper aims to innovate and provide complementary results to the main topic. Abstract is well written although authors may consider some suggestions. The introduction is lacking of some points and besides it needs to be reorganized. Materials and methods are very complete, only a few details are missing. Results are precise and concise. Please see my comments below. 

ABSTRACT

L12. Authors may want to add a brief introductory sentence of the paper, instead of starting with the MM directly. Also, a brief description of the use/purpose of EAWP would enrich the abstract.

L16. Please specify the organism of the genes before the name of the gene, FiCHI or FiGLU (italics), and check the correct spelling for genes.

L16. Please add: … genes of F. incarnatum.

L22. Authors may want to consider: The results of SEM microscopy revealed that the effect of the crude enzymes of EAWP-treated muskmelon caused morphology changes in F. incarnatum mycelia.

L26. Please, try to use the same concepts. Please use “untreated” instead of “control”. Or, specify from the beginning that “untreated (control)”.

INTRODUCTION

L37. Do you have any number of the annual production, or % of increase?

L43. Please replace “the pathogen” with causal agent, pathogenic agent…

L43. Please, add brief information on how this pathogen infects the host, through wounds, machinery… Also, introduce the PR proteins here since they are a very important part of the results.

L44. Two times Abiotic. Please revise.

L44-45. Plants produce ROS among other responses. ROS is not the only one. Please, revise.

From my point of view, the 3 last paragraphs of the introduction are disorganized. I think that after the introduction the pathogen (L43), the next sentences could be L56, then L44-48, L57-59, introduction to EAWPs as postharvest treatment (it’s missing), L50, L60/62. Please consider this reorganization or another similar to it.

MATERIALS AND METHODS

L74. Muskmelons were immediately subjected to pathogen inoculation. Was the fruit dried or wet?

L85. Conidia were collected

L86. DW is distilled water? Please revise. Did the authors use water without tween or any other surfactant? Why?

L87. Please add the size of the needle.

L96. Which modifications?

L97. As far as I see, it is the first time for the PR abbreviation. Please clarify.

L104. 1ng of cDNA, not RNA

L108-110. Please, re-write. Also, consider that actin is the internal reference gene, a basal gene, not “the control”.

L107. Are these primers designed de novo, or are they from literature? Please clarify.

L127. SEM conducted on the peels or agar plugs?

L154. When did the authors use t-test? When Tukey’s test?

RESULTS

L162. Add figure 2 here.

L180. This is discussion, not results.

Section 3.2 These results are expected because there is no pathogen on the untreated samples. So, authors may want to clarify why they did this, either here or in the discussion. Relate this to the activity of the pathogen when invading a host etc. Also, this would be more interesting if the authors should have the data along the 3 time points after inoculation. It would enrich the manuscript.

L209. This goes to the discussion. Also, the authors may want to add the characteristics of the pathogen cells before, i.e., introduction?

Please, apply these changes and revise the discussion with my previous comments since some of them also apply to it. 

Author Response

Reviewer 1

This paper aims to innovate and provide complementary results to the main topic. Abstract is well written although authors may consider some suggestions. The introduction is lacking of some points and besides it needs to be reorganized. Materials and methods are very complete, only a few details are missing. Results are precise and concise. Please see my comments below. 

Answer: Thank you for your review and valuable suggestion to improve this manuscript.

ABSTRACT

L12. Authors may want to add a brief introductory sentence of the paper, instead of starting with the MM directly. Also, a brief description of the use/purpose of EAWP would enrich the abstract.

Answer: We have added brief introduction and brief detail of EAWPs in the first and second paragraph of abstract as “Postharvest quality of muskmelon is faced with fruit rot caused by a fungus Fusarium incarnatum in relation to the loss of quality. Utilization of electrostatic atomized water particles (EAWPs) in agriculture application has been shown to induce disease resistance in plants. Therefore, in this study, we determined…

L16. Please specify the organism of the genes before the name of the gene, FiCHI or FiGLU (italics), and check the correct spelling for genes.

Answer: In this case, expression of gene was conducted in muskmelon tissues, therefore we have revised as CmCHI and CmGLU.

L16. Please add: … genes of F. incarnatum.

Answer: According to expression of gene was conducted in muskmelon tissues, therefore we have added “genes of Cucumis melo” into abstract.

L22. Authors may want to consider: The results of SEM microscopy revealed that the effect of the crude enzymes of EAWP-treated muskmelon caused morphology changes in F. incarnatum mycelia.

Answer: We have revised as suggestion.

L26. Please, try to use the same concepts. Please use “untreated” instead of “control”. Or, specify from the beginning that “untreated (control)”.

Answer: We have used “untreated” instead of control throughout abstract.

INTRODUCTION

L37. Do you have any number of the annual production, or % of increase?

Answer: We have added annual production.

L43. Please replace “the pathogen” with causal agent, pathogenic agent…

Answer: We have replaced with “causal agent”.

L43. Please, add brief information on how this pathogen infects the host, through wounds, machinery… Also, introduce the PR proteins here since they are a very important part of the results.

Answer: We have added brief information of pathogen infects host and introduction of PR proteins in introduction parts as “Fusarium sp. normally infects plants through wounds during rain or irrigation [5]. Pathogen starts to invade plant tissues and rapidly spread to other parts of plants. General morphology characteristics of F. incarnatum represented cotton-like colony and rapid growth on PDA. Hyphae of F. incarnatum were hyaline septate with smooth cell wall [5]. Some Fusarium isolates produce cellulase and pectinase contribute to weakening and invading the host plant successfully [6], which caused tissue rot.”

L44. Two times Abiotic. Please revise.

Answer: We have revised as abiotic and biotic stresses.

L44-45. Plants produce ROS among other responses. ROS is not the only one. Please, revise.

Answer: We have added and rearrangement of this paragraph “Plants monitor a broad range of defense mechanisms against abiotic and biotic stresses. Plants respond to abiotic and biotic stresses by oxidative burst of cells, synthesis of phytoalexin and production of pathogenesis related proteins (PR) and producing and accumulating reactive oxygen species (ROS) [7–9].”

From my point of view, the 3 last paragraphs of the introduction are disorganized. I think that after the introduction the pathogen (L43), the next sentences could be L56, then L44-48, L57-59, introduction to EAWPs as postharvest treatment (it’s missing), L50, L60/62. Please consider this reorganization or another similar to it.

Answer: We have rearrangement according to reviewer comments and add introduction of EAWPs in postharvest treatment.

MATERIALS AND METHODS

L74. Muskmelons were immediately subjected to pathogen inoculation. Was the fruit dried or wet?

Answer: We used fresh fruit (wet) for inoculation in this experiment.

L85. Conidia were collected

Answer: We have revised as “conidia of F. incarnatum were collected”

L86. DW is distilled water? Please revise. Did the authors use water without tween or any other surfactant? Why?

Answer: We have revised as distilled water (DW). In this experiment we did not use tween or other surfactant for inoculation due to point inoculation at wounded tissues.

L87. Please add the size of the needle.

Answer: We have added “needle (0.8 mm in diameter)”.

L96. Which modifications?

Answer: For this method, we did not extract plant tissue to test total hydrogen peroxide content, but this study we used distilled water to receive hydrogen peroxide released from EAWPs.

L97. As far as I see, it is the first time for the PR abbreviation. Please clarify.

Answer: We have added detail about PR protein in introduction part.

L104. 1ng of cDNA, not RNA

Answer: We have revised as cDNA.

L108-110. Please, re-write. Also, consider that actin is the internal reference gene, a basal gene, not “the control”.

Answer: We have revised as suggestion.

L107. Are these primers designed de novo, or are they from literature? Please clarify.

Answer: Primers were designed from Primer3.

L127. SEM conducted on the peels or agar plugs?

Answer: SEM were conducted on agar plug of F. incarnatum that incubated in crude metabolites extracted from EAWPs treated muskmelon peel.

L154. When did the authors use t-test? When Tukey’s test?

Answer: Student t-test was used to compare between 2 treatment (relative gene expression and enzyme assay) and Tukey’s test was used to compare treatments more than 2 treatments (lesion development).

RESULTS

L162. Add figure 2 here.

Answer: We have revised as suggestion.

L180. This is discussion, not results.

Answer: We have moved to discussion part.

Section 3.2 These results are expected because there is no pathogen on the untreated samples. So, authors may want to clarify why they did this, either here or in the discussion. Relate this to the activity of the pathogen when invading a host etc. Also, this would be more interesting if the authors should have the data along the 3 time points after inoculation. It would enrich the manuscript.

Answer: We have discussed about this phenomenon in discussion part as “Although, expression of PR genes (CmCHI and CmGLU) and enzymes activity of chitinase and β-1,3-glucanase in EAWPs untreated muskmelon fruits demonstrated significantly lower expression and activity level compared to EAWPs treated muskmelon. However, untreated muskmelon fruits showed expression of PR genes and activity of both enzymes. This phenomenon may be due to PR proteins are strongly induced by wounding or infection by pathogens which have been reported that wounding plant tissues induced expression of PR genes in tobacco [39]. Furthermore, artificial wounding has been reported enhanced expression level of PR-10 in bilberry leaf [40]. Our study, muskmelon fruits were wounded by needles before inoculation by F. incarnatum spore suspension and DW (control). This wounding may induce expression of PR genes and enzymes activity as observed in Figs 4 and 5.”

L209. This goes to the discussion. Also, the authors may want to add the characteristics of the pathogen cells before, i.e., introduction?

Answer: We have moved these sentence to discussion part. General characteristic of F. incarntum was added into introduction as “General morphology characteristics of F. incarnatum represented cotton-like colony and rapid growth on PDA. Hyphae of F. incarnatum were hyaline septate with smooth cell wall [5].”

Please, apply these changes and revise the discussion with my previous comments since some of them also apply to it. 

Answer: Any changes by revision were indicated by red text throughout this manuscript.

Reviewer 2 Report

The manuscript titled “Electrostatic atomized water particles Induce disease resistance in muskmelon (Cucumis melo L.) against postharvest fruit rot caused by Fusarium incarnatum” by Samak Kaewsuksaeng et al. describes the effect of pretreatment with electrostatic atomized water particles (EAWPs) in muskmelon on production of H2O2, disease progress, induction of defense-related gene expression (chi and glu) and activities of cell wall degrading enzymes (chitinase and β-1,3-glucanase), morphology of pathogen F. incarnatum, and maintenance of post-harvest muskmelon quality.

Before accepting this article, following questions need to be addressed:

1. Muskmelon fruits were inoculated with Fusarium incarnatum after treatment with EAWPs for 0 (control), 30, 60, 90, or 120 minutes. Figure 2 presents H2O2 accumulation for all EAWPs treatments, and similarly figure 3 shows postharvest fruit rot development for all EAWPs treatments. Why the authors didn’t present gene expression and cell wall degrading enzymes results for 60, 90 and 120 minutes EAWPs treatments? At least the results associates with those treatments (60, 90, 120 min) should have been included in the discussion section.

2. The authors state that “This finding suggested that the H2O2 generated by EAWPs maintained fruit firmness and color in muskmelon fruits.”  The figure 1 and figure 2 results does not support that conclusion.

3. Page 3 – section 2.4. qRT-PCR of PR Protein Genes, Line 99: EAWP-treated muskmelons were subjected to RNA extraction using Trizol reagent (ThermoFisher, Waltham, MA, USA), and total RNA was dissolved in … Not clear whether a part of the fruit was used or the whole fruit was used for RNA extraction?

Author Response

Reviewer 2

The manuscript titled “Electrostatic atomized water particles Induce disease resistance in muskmelon (Cucumis melo L.) against postharvest fruit rot caused by Fusarium incarnatum” by Samak Kaewsuksaeng et al. describes the effect of pretreatment with electrostatic atomized water particles (EAWPs) in muskmelon on production of H2O2, disease progress, induction of defense-related gene expression (chi and glu) and activities of cell wall degrading enzymes (chitinase and β-1,3-glucanase), morphology of pathogen F. incarnatum, and maintenance of post-harvest muskmelon quality.

Answer: Thank you your review and valuable suggestion to improve this manuscript.

Before accepting this article, following questions need to be addressed:

  1. Muskmelon fruits were inoculated with Fusarium incarnatumafter treatment with EAWPs for 0 (control), 30, 60, 90, or 120 minutes. Figure 2 presents H2O2 accumulation for all EAWPs treatments, and similarly figure 3 shows postharvest fruit rot development for all EAWPs treatments. Why the authors didn’t present gene expression and cell wall degrading enzymes results for 60, 90 and 120 minutes EAWPs treatments? At least the results associates with those treatments (60, 90, 120 min) should have been included in the discussion section.

Answer: Thank you for this concern about relative gene expression and enzyme activity of EAWPs treated muskmelon. Actually, in this study, we have tested an appropriate time for application onto muskmelon fruits by varying times, and observed response of muskmelon against F. incarnatum inoculation by phenotypic expression. Based on our result, application for 30 min showed the most appropriate time for limit fungal invasion and symptom development. Therefore, we selected result for 30 min for further study. We have included this phenomenon in discussion as “We selected EAWPs application time for 30 min to investigate gene expression and enzyme activities in muskmelon fruits against F. incarnatum due to capacity to limit lesion development (Fig. 3).”

  1. The authors state that “This finding suggested that the H2O2 generated by EAWPs maintained fruit firmness and color in muskmelon fruits.”  The figure 1 and figure 2 results does not support that conclusion.

Answer: The conclusion may related to results of Fig. 7 that revealed fruit stiffness (firmness) and chlorophyll content. We have revised as “This finding suggested that the H2O2 generated by EAWPs maintained fruit firmness and chlorophyll content in muskmelon fruits”.

  1. Page 3 – section 2.4. qRT-PCR of PR Protein Genes, Line 99: EAWP-treated muskmelons were subjected to RNA extraction using Trizol reagent (ThermoFisher, Waltham, MA, USA), and total RNA was dissolved in … Not clear whether a part of the fruit was used or the whole fruit was used for RNA extraction?

Answer: Total RNA was extracted from muskmelon peels, therefore we have revised as “EAWP-treated muskmelon peels were subjected to RNA extraction…”

Reviewer 3 Report

(Line 32) Introduction section: taking into account that this technique has been previously evaluated in several scientific studies, the introduction section lacks a description of similar studies that have allowed EAWP treatment to be selected for the control of the disease in melon.

(Line 32) Introduction section: It should be useful to describe the usual storage conditions of Muskmelons that could be treated with EAWP and the economic impact it could have, both for its application and for disease control.

(Line 32). Introduction section: Describe the incidence, severity and progression of the disease as well as the sources of inoculum and transmission of the fungus under storage conditions.

(Line 79) Pathogen inoculation: Does the methodology for pathogen inoculation correspond to or is it representative of the epidemiology of the disease under real conditions?

(Line. 81) It is necessary to describe the pathogen: Origin of the pathogen isolate and how the Faculty of Natural Resources identified the fungal specie. Was the isolate isolated from Muskmelon?

Taking into account the complex taxonomy of the genus Fusarium, was the inoculation carried out with monosporic isolates?

(Line 96). “method with some modifications”, These should be specified.

Figure 3: Describe the meaning of the error bars shown and write under the figure the given parameter (min.).

Figure 5: Describe the meaning of the error bars.

Line 225. Discussion section. It would be useful to discuss the full-scale applicability of this disease control methodology under actual Muskmelon storage conditions..

Author Response

Reviewer 3

(Line 32) Introduction section: taking into account that this technique has been previously evaluated in several scientific studies, the introduction section lacks a description of similar studies that have allowed EAWP treatment to be selected for the control of the disease in melon.

Answer: We have described utilization of EAWPs to maintain postharvest quality in several plants and for disease control in tomato as “For postharvest quality, application of EAWPs with appropriate times (90 min) maintained color and total chlorophyll content of the spar of asparagus during storage at 4°C [21]. Furthermore, utilization of EAWPs for 60 min maintained postharvest quality of ‘Namwa’ banana through delay ripening and senescence [19]. Furthermore, for disease control application of EAWP has been shown to induce disease resistance in tomato by upregulated nitrate reductase gene and salicylic-dependent chitinase 3 [22].”

(Line 32) Introduction section: It should be useful to describe the usual storage conditions of Muskmelons that could be treated with EAWP and the economic impact it could have, both for its application and for disease control.

Answer: We have added in the last paragraph of introduction as “According to postharvest fruit rot negatively impacted for quality and quantity of muskmelon fruit production with 10% disease incidence in market shelf. Pretreatment of EAWPs with appropriate time will help to decrease disease development and to maintain postharvest quality of muskmelon fruit during storage time.”

(Line 32). Introduction section: Describe the incidence, severity and progression of the disease as well as the sources of inoculum and transmission of the fungus under storage conditions.

Answer: We have added in introduction as “During the rainy season about 10% of muskmelon fruit were infected with fruit rot and postharvest fruit rot [5]. The disease primarily started from wounded or cracked muskmelon fruits near peduncles, then became rotten with mycelia covered lesion, wet rotten tissues were observed beneath the lesion [5].”

(Line 79) Pathogen inoculation: Does the methodology for pathogen inoculation correspond to or is it representative of the epidemiology of the disease under real conditions?

Answer: Fusarium incarnatum is airborne pathogen which can be spread and transmission by spore. For inoculation in this study, we used spore suspension to inoculate onto muskmelon fruits. This method considered as artificial inoculation similar as several previous researches. 

(Line. 81) It is necessary to describe the pathogen: Origin of the pathogen isolate and how the Faculty of Natural Resources identified the fungal specie. Was the isolate isolated from Muskmelon?

Answer: Actually the disease pathogen was identified and published in previous study and collected in culture collection of our faculty. Therefore, we revised as “The postharvest fruit rot pathogen of muskmelon, F. incarnatum which isolated from infected muskmelon in previous study [5], was obtained from the Culture Collection of Pest Management, Faculty of Natural Resources, Prince of Songkla University, Thailand.

Taking into account the complex taxonomy of the genus Fusarium, was the inoculation carried out with monosporic isolates?

Answer: Actually, F. incarnatum was cultured from the stock which collected in Culture Collection. Before inoculation onto muskmelon we have to confirm its morphology and all cultured should be the carried out with monosporic isolates.

(Line 96). “method with some modifications”, These should be specified.

Answer: For this method, we did not extract plant tissue to test total hydrogen peroxide content, but this study we used distilled water to receive hydrogen peroxide released from EAWPs.

Figure 3: Describe the meaning of the error bars shown and write under the figure the given parameter (min.).

Answer: We have described error bar as “Error bars indicate standard deviation (SD) of application time (min.) whereas letters indicate significant difference among treatments according to Tukey’s test (p < 0.05).”

Figure 5: Describe the meaning of the error bars.

Answer: Error bars indicate standard deviation (SD) of enzyme activity (unit) whereas letters indicate significant differences among treatments according to Tukey’s test (p < 0.05).

Line 225. Discussion section. It would be useful to discuss the full-scale applicability of this disease control methodology under actual Muskmelon storage conditions.

Answer: We have added in discussion as “Based on results from the present study, pretreatment of muskmelon fruits by EAWPs for 30 min induced disease resistance against F. incarnatum and limited lesion development. In order to prevent muskmelon fruits from postharvest fruit rot and to maintain postharvest quality. Pretreatment with EAWPs for 30 min would recommend to apply on muskmelon after harvesting and/or before deliver to market shelf.”